# Association of SARS-CoV-2 vaccination status with risk of influenza-like illness and loss of workdays in healthcare workers

Tamara Dörr [1,16], Joanne Lacy[2,16], Tala Ballouz [2], Alexia Cusini[3], Fabian Grässli[1], Sarah Haile[2], Emina Kocan[4], J. Carsten Möller[5], Milo A. Puhan[2], Matthias Schlegel[1], Matthias von Kietzell[6], Markus Rütti[7], Reto Stocker [8], Danielle Vuichard Gysin[9,10], Christian R. Kahlert [1,11], Stefan P. Kuster[1], Philipp Kohler [1,2] ✉ & for the SURPRISE+ Study Group*

## Abstract

**Background** In the post-pandemic phase, the value of annual SARS-CoV-2 booster vaccination in healthcare-workers is unclear. In this multicentre cohort study, we sought to determine the association of SARS-CoV-2 vaccination status and other risk factors with the occurrence of influenza-like respiratory illness and workdays lost due to influenza-like respiratory illness.

**Methods** During a period of high SARS-CoV-2 community transmission (November 2023 to May 2024), we collected weekly data on symptoms and sick day leave and used negative binomial regression to identify risk factors for these outcomes among 1745 healthcare workers. To single out the effect of the vaccine and account for potential confounding, additional inverse probability weighted analysis was performed.

**Results** In both analyses, we show that more SARS-CoV-2 vaccinations are associated with a higher risk of influenza-like respiratory illness and workdays lost. For influenza-like respiratory illness, the association is stronger with a more recent timing of the vaccination rather than the number of vaccinations, which suggests that the effect wanes over time. In contrast, seasonal influenza vaccination is associated with a decreased risk for both outcomes.

**Conclusions** Based on our data, we conclude that SARS-CoV-2 booster vaccination does not contribute to the protection of the healthcare workforce in a post-pandemic setting. SARS-CoV-2 vaccination may even temporarily increase the likelihood of symptomatic infection and workday loss.

## Plain language summary

In the years following the COVID-19 pandemic, it is still unclear whether annual COVID booster vaccines are necessary for low-risk populations such as healthcare workers. In this study, 1745 healthcare workers in Switzerland were followed over several months to see how their vaccination status affected their chances of getting flu-like illnesses and missing work. The study found that those who recently received a COVID-19 booster were more likely to report symptoms and take sick leave. In contrast, people who received the seasonal flu vaccine were less likely to report symptoms or miss work. These findings suggest that COVID-19 boosters may not offer clear short-term benefits in a post-pandemic setting, and may even increase short-term illness risk. This raises questions about the best use of booster vaccines moving forward.

The SARS-CoV-2 vaccine has contributed majorly towards ending the pandemic worldwide and booster doses have been recommended for populations at risk[1,2]. Healthcare workers (HCW), being at the forefront of the pandemic, were initially considered as a population at risk due to high exposure to infectious patients. In the current transition to SARS-CoV-2 endemicity, the value of renewed vaccination for this population is under debate as protection against the currently circulating SARS-CoV-2 viral strains is unclear and COVID-19 is usually a mild disease in young to middle-aged, healthy individuals. Also, potential adverse effects of the vaccination need to be taken into consideration. Immediate reactions after vaccination such as injection site pain, headache, or fever, are common and, while generally benign, can be of debilitating severity[3]. Recent observational studies have also suggested a higher risk of reinfection after booster vaccination[4–9] and while this might be attributable to selection bias[7,10], careful consideration of risk and benefits is warranted. On the other hand, the successful prevention of infection could preserve HCW workforce as a

A full list of affiliations appears at the end of the paper. *A list of authors and their affiliations appears at the end of the paper. ✉e-mail: philipp.kohler@h-och.ch

valuable societal resource and – via prevention of transmission – also protect patients.

In a cohort of HCW, we sought to identify factors associated with the occurrence of influenza-like respiratory illness (ILI) episodes and workdays lost due to ILI during a period of high SARS-CoV-2 community transmission. In particular, we aimed to single out the impact of SARS-CoV-2 vaccination status on these outcomes. The results suggest that in a HCW population, SARS-CoV-2 booster vaccination is associated with a temporarily elevated risk for ILI and workdays lost, whereas influenza vaccination is associated with a decreased risk.

## Methods
### Setting and participants
In our prospective multicentre HCW cohort, participants were recruited from nine healthcare networks in Northern and Eastern Switzerland and followed since 08/2020. All hospital employees with or without patient contact, aged 16 years or older, were eligible for inclusion and enroled upon provision of informed consent. The study and protocol were approved by the Ethics Committee of Eastern Switzerland (#2020–00502).

### Data collection
In October 2023, all new and ongoing participants provided or updated their baseline data (i.e., age, sex, health determinants, occupational and social life factors) and details on their history of SARS-CoV-2 infections (i.e., number and date of positive test results) and SARS-CoV-2 vaccinations (i.e., number, date and type). Participants were also asked to provide a serum sample tested for SARS-CoV-2 anti-spike (anti-S) and anti-nucleosid (anti-N) antibodies. Anti-S and anti-N were detected with the Roche Elecsys (Roche Diagnostics, Rotkreuz, Switzerland) electro-chemiluminescence immunoassay[11]. In weekly follow-up questionnaires between November 1st 2023 and April 30th 2024, participants indicated the presence of any of 22 respiratory, gastrointestinal and general symptoms (Supplementary Table 1) with an acute onset (new occurrence in the preceding 7 days) during the last 7 days, days of work absence attributable to symptoms, and documented any vaccination against SARS-CoV-2 or seasonal influenza including details on type of vaccine.

For this analysis, we included only those 1745 (87.6%) participants who provided at least 50% of follow-up questionnaires (i.e., 13 or more). Furthermore, we excluded those reporting more than 4 vaccinations, as ≥5

doses were only recommended for highly selected populations by the Swiss government, and those receiving a SARS-CoV-2 vaccination during the follow-up period.

### Predictors and outcomes
Vaccination status was used as main predictor and treated as categorical variable, as we assumed a non-linear effect of the number of vaccinations on the outcome. Participants were allocated to being unvaccinated; having received 1 or 2 SARS-CoV-2 vaccine doses (because after the first COVID-19 wave, people with one vaccine dose and documented infection were considered fully vaccinated), 3 vaccinations (majority with 1st booster); or 4 vaccinations (majority with 2nd booster). Of those with 4 vaccinations, 85.2% received the bivalent vaccine. Exact definitions of other predictors are shown in Supplementary Table 2.

ILI was chosen as main outcome as it can indicate SARS-CoV-2 activity during periods of high community transmission levels[12,13] and testing for SARS-CoV-2 has been widely abolished in the post-pandemic phase. During the study period, 21.1% of tested individuals in Switzerland with ILI were SARS-CoV-2 positive and 20.0% were positive for influenza[14]. In accordance with the Centres for Disease Control and Prevention (CDC)[15] and European Centre for Disease Prevention and Control (ECDC)[16], ILI was defined as the occurrence of fever (≥38.0 °C) or feeling of feverishness AND a respiratory symptom (cough, sore throat, rhinitis or the loss of smell) AND an acute onset ≤7 days before respective reporting date. Sensitivity analyses were performed using two different case definitions, one being more lenient (acute onset of fever ≥ 38.0 °C or feeling of feverishness AND any other of the symptoms asked), one being more restrictive (fever ≥ 38.0 °C or feeling of feverishness AND ≥ 1 general symptom among fatigue, headache, and malaise AND ≥ 2 other symptoms). As secondary outcome, the number of workdays lost due to these symptoms was examined.

### Statistical analyses
Baseline characteristics and outcomes by vaccination status were compared using two-sided Chi-square tests for categorical variables and two-sided Kruskal-Wallis test for continuous variables (assuming non-normal distribution).

To identify factors associated with number of ILI and workdays lost, uni- and multivariable regression analysis were performed using negative binomial models with number of answered follow-up questionnaires as

**Fig. 1 |** Flowchart of participants including reasons for study exclusion and sample sizes by vaccination status.

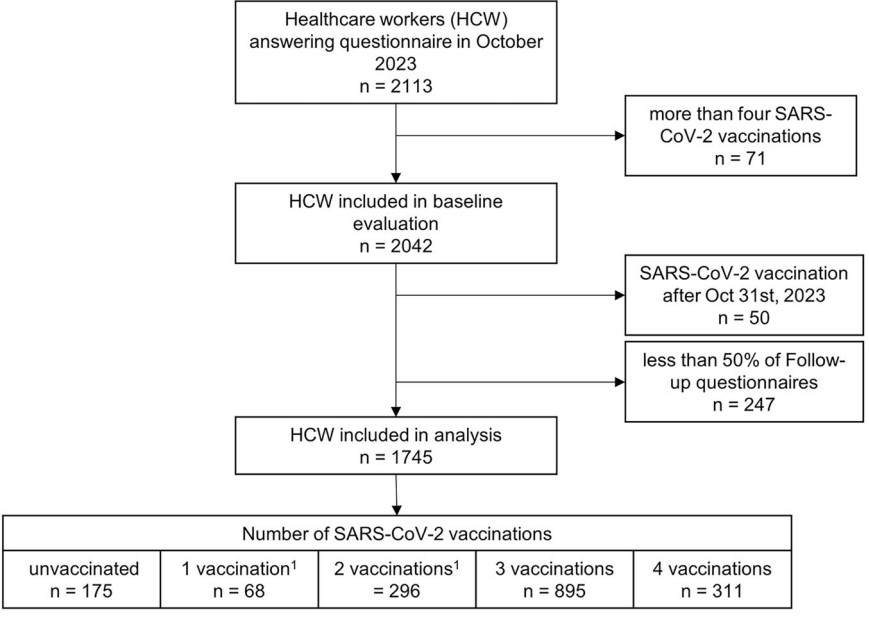

¹evaluated as one group for the analysis

**Table 1 | Baseline HCW characteristics in October 2023 (*n* and % unless mentioned otherwise)**

| | Unvaccinated | 1 or 2 doses | 3 doses | 4 doses | *p-value*[1] |
|---|---|---|---|---|---|
| *n* | 175 | 364 | 895 | 311 | |
| Characteristics | | | | | |
| Age, median (range) | 45 (17–63) | 44 (18–70) | 48 (19–68) | 50 (24–68) | 8.42e−10 |
| Sex = males | 24 (13.7) | 58 (15.9) | 162 (18.1) | 77 (24.8) | 0.006 |
| BMI[2], median (IQR[3]) | 24 (22–27) | 24 (21–26) | 23 (21–26) | 24 (22–27) | 0.385 |
| Active smoking | 28 (16.0) | 51 (14.0) | 111 (12.4) | 50 (16.1) | 0.313 |
| Any comorbidity | 21 (12.0) | 63 (17.3) | 181 (20.2) | 80 (25.7) | 0.002 |
| Hypertension | 5 (3.0) | 20 (5.8) | 68 (7.9) | 35 (11.5) | 0.004 |
| Diabetes mellitus | 1 (0.6) | 4 (1.2) | 11 (1.3) | 4 (1.3) | 0.900 |
| Cancer | 1 (0.6) | 4 (1.2) | 17 (2.0) | 4 (1.3) | 0.488 |
| Immunosuppressed | 4 (2.4) | 6 (1.7) | 38 (4.4) | 8 (2.6) | 0.083 |
| Cardiovascular | 2 (1.2) | 5 (1.5) | 15 (1.7) | 12 (3.9) | 0.069 |
| Pulmonary | 8 (4.8) | 29 (8.5) | 51 (5.9) | 31 (10.2) | 0.037 |
| Profession | | | | | 2.65e−11 |
| Physician | 11 (6.3) | 19 (5.2) | 114 (12.7) | 70 (22.5) | |
| Nurse | 96 (54.9) | 203 (55.8) | 386 (43.1) | 117 (37.6) | |
| Therapist | 12 (6.9) | 20 (5.5) | 38 (4.2) | 22 (7.1) | |
| Administration | 22 (12.6) | 52 (14.3) | 167 (18.7) | 50 (16.1) | |
| Other | 34 (19.4) | 70 (19.2) | 190 (21.2) | 52 (16.7) | |
| Patient contact | 129 (73.7) | 251 (69.0) | 611 (68.3) | 220 (70.7) | 0.499 |
| Influenza vaccine 2023/24 | 8 (4.6) | 50 (13.7) | 236 (26.4) | 156 (50.2) | 1.26e−35 |
| SARS-CoV-2 infections, median (range) | 1 (0–2) | 1 (0–3) | 1 (0–4) | 1 (0–4) | 1.44e−16 |
| Time of last vaccination | | | | | 4.25e−304 |
| before 11/21 | N/A | 210 (57.7) | 10 (1.1) | 0 (0.0) | |
| 11/21 to 09/22 | N/A | 145 (39.8) | 852 (95.2) | 18 (5.8) | |
| after 09/22 | N/A | 9 (2.5) | 33 (3.7) | 293 (94.2) | |
| ≥1 bivalent vaccine | N/A | 16 (4.4) | 38 (4.2) | 265 (85.2) | 4.05e−246 |
| SARS-CoV-2 serology (n) | 132 | 278 | 681 | 238 | |
| Anti-N-positive (%)[4] | 128 (97.0) | 262 (94.2) | 614 (90.2) | 202 (84.9) | 1.14e−04 |
| Anti-S-titre (BAU/mL), median (IQR[3])[5] | 537 (93-1811) | 5000 (3808-5000) | 5000 (5000-5000) | 5000 (5000-5000) | 8.06e−102 |

[1]Chi-square test for categorical variables, Kruskal-wallis for continuous variables; [2]*BMI* Body Mass Index (in kg/m2), [3]*IQR* Interquartile range; [4]positive defined as cut-off-index>0.1; [5]*BAU* Binding Antibody Units.

offset term (complete case analysis). Incidence rate ratios (IRR) with corresponding 95% confidence intervals (CI) were calculated with adjustment for a priori defined confounders selected based on scientific knowledge of risk factors and associations found earlier in our cohort. These were age, sex, body mass index (BMI), smoking status, presence of any relevant comorbidity (i.e., cancer, immunosuppressive disorders, cardiovascular disease, lung disease), living with children under the age of 12, total number of positive SARS-CoV-2 swabs reported since the beginning of the pandemic and until October 2023, patient contact, and receipt of the seasonal influenza vaccine for 2023/2024 (Supplementary Table 2). Because of suspected multicollinearity of vaccination status and time of last vaccination, two different models (model 1: without time of last vaccination; model 2: with time of last vaccination) were fitted. To account for influenza vaccination status as the potentially most important confounder, subgroup analysis for the outcome ILI was performed, including only HCW without seasonal influenza vaccination. To investigate the effect of bivalent vaccine formulations, an additional sensitivity analysis was performed, where participants receiving either 3 or 4 vaccinations were grouped together to avoid multi-collinearity (as those with bivalent vaccines were mostly those receiving 4 vaccinations).

To test the robustness of our findings and reduce potential confounding in investigating the effect of the SARS-CoV-2 vaccine, we performed inverse probability of treatment weighting (IPTW). First, propensity scores were calculated using generalized boosted model regression (mnps function from

the R package 'twang') with the number of vaccines served as outcome and the following as independent variables which are known to influence the SARS-CoV-2 infection risk: age, sex, BMI, comorbidities, patient contact, children at home, previous positive swabs, and smoking status. To account for extreme propensity scores and improve robustness, the overlap weighting method[17] was used to calculate the weights. Covariate balance after weighting was assessed using standardized mean differences (SMDs) with SMDs of less than 0.1 indicating sufficient balance. In the IPTW analysis, both negative binomial models were performed, which allowed for a more accurate estimation of the average treatment effect of receiving a certain number of vaccines on the number of ILIs. We used statistical software R (version 4.4.0) with the packages 'tableone', 'nlme', 'MASS' and 'twang' for the analyses. Statistical significance level was defined at α = 0.05.

## Reporting summary

Further information on research design is available in the Nature Portfolio Reporting Summary linked to this article.

## Results
### Population

Of 2113 HCW who filled in the baseline questionnaire in September 2023, 1745 (82.6%) were included, with a median age of 46 years (range 17–70) and 81.6% being female (Fig. 1). Of these, 175 (10.0%) were unvaccinated,

**Fig. 2 | Participants experiencing ILI episodes and workdays lost during follow-up per vaccination group. a** Percentage of participants (*n* = 1745) experiencing influenza-like illness (ILI) episodes during follow-up, stratified by vaccination group. Incidence Rate Ratios (IRRs) with 95% Confidence Intervals from univariable negative binomial regression are shown, using the unvaccinated group as the reference. **b** Percentage of participants (*n* = 1745) reporting workdays lost during follow-up, stratified by vaccination group. IRRs with 95% Confidence Intervals from univariable negative binomial regression are shown, with the unvaccinated group as the reference. *P*-values are derived from the Wald test.

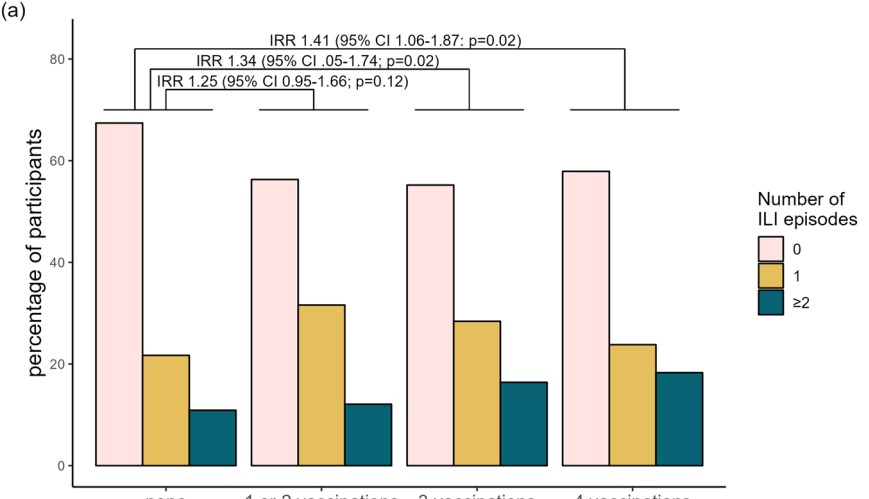

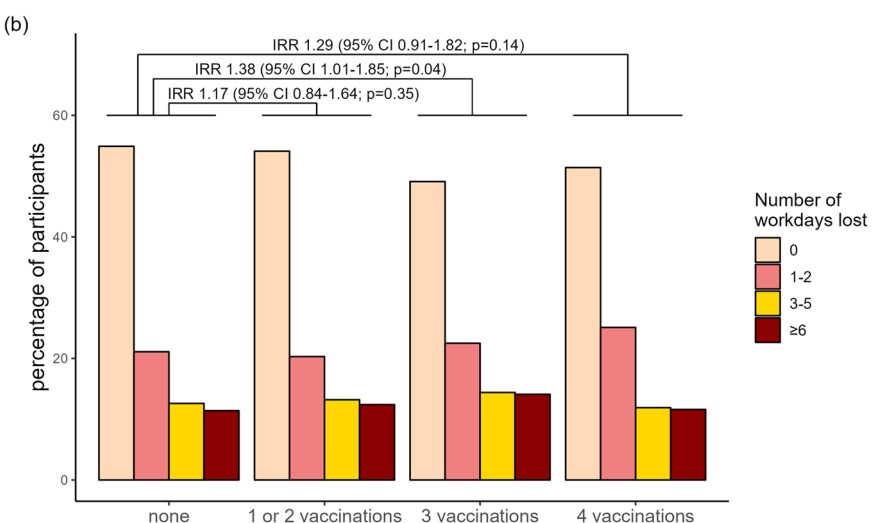

364 (20.9%) had received 1 or 2 vaccinations, 895 (51.3%) reported 3, and 311 (17.8%) reported 4 vaccine doses. Those with 3 or 4 SARS-CoV-2 vaccinations tended to be older, more likely to report comorbidities and having received seasonal influenza vaccination. They also reported fewer previous SARS-CoV-2 infections and were less likely to have detectable anti-nucleosid (anti-N) antibodies (Table 1). Of SARS-CoV-2 vaccinated individuals, 1'534 (97.7%) had received ≥1 vaccine based on messenger RNA (mRNA) technology and 319 (18.3%) had received ≥1 booster with a bivalent vaccine (Supplementary Fig. 1). Of those with 4 vaccine doses, 265 (85.2%) had received the bivalent vaccine compared to 54 (4.3%) of those with less than 4 doses.

### *Epidemiological context and univariable analysis of outcomes*
The frequency distribution of ILI - defined according to the CDC and ECDC - followed the local epidemiology of SARS-CoV-2 and influenza in the general population (Supplementary Fig. 2), with 748 (42.9%) HCW reporting at least one ILI during follow-up and 35.7% (267/748) experiencing more than one episode. Overall, 853 (48.9%) reported at least one day of absence due to ILI symptoms, with the majority (76.6%) being on sick leave for several days. In univariable analysis, vaccination status with 3 and 4 doses was significantly associated with the number of ILI (IRR 1.34, 95% CI 1.05–1.74 and 1.41, 95% CI 1.06–1.87 respectively), and, for 3 doses also with workdays lost (IRR 1.38, 95% CI 1.01–1.85), (Fig. 2; for raw numbers, see supplementary Table 3).

### Multivariable analysis of risk factors associated with ILI
In multivariable (negative binomial) regression model 1 (without the variable time of last SARS-CoV-2 vaccination), the risk of ILI was significantly associated with vaccination status for those with 3 or 4 vaccinations (aIRR 1.56, 95% CI 1.22–2.03 and 1.70, 95% CI 1.27–2.28, respectively). Also, number of positive SARS-CoV-2 tests and the presence of comorbidities were positively associated with number of ILI episodes, while seasonal influenza vaccination and age were associated with a decreased risk. Upon inclusion of the timing of last vaccination (categorical variable based on the recommendations of the Swiss government in model 2), the risk of ILI was no longer associated with the number of vaccinations, but more recent vaccination was significantly associated with the number of ILI (aIRR 1.32, 95% CI 1.07–1.62). For the other variables, similar results were found as in model 1 (Table 2). Sensitivity analyses with both more lenient and more restrictive definitions of ILI showed the same trends (Supplementary Tables 4, 5) as did a sensitivity analysis evaluating the effect of bivalent vaccine formulations (Supplementary Table 6). Restricting the analysis to those without influenza vaccination did not show any differences, either (Supplementary Table 7).

### Inverse probability of treatment weighting (IPTW)
To test the robustness of our findings, we performed a sensitivity analysis using IPTW which further accounts for potential confounding by balancing participant characteristics across categories of vaccination status. After applying overlap weighting, all covariates included in the models

**Table 2 | Multivariable regression analysis for factors associated with number of ILI episodes as outcome**

| Characteristics | Model 1: Without time | | | Model 2: With time | | |
|---|---|---|---|---|---|---|
| | aIRR[1] | 95% CI[1] | *p*-value | aIRR[1] | 95% CI[1] | *p*-value[2] |
| Time of last Vaccine | | | | 1.32 | 1.07,1.63 | 0.009 |
| Number of Vaccines | | | | | | |
| 0 | — | — | | — | — | |
| 1 or 2 | 1.29 | 0.98,1.71 | 0.069 | 0.85 | 0.56,1.30 | 0.451 |
| 3 | 1.56 | 1.22,2.03 | 5.63e−04 | 0.89 | 0.54,1.46 | 0.641 |
| 4 | 1.70 | 1.27,2.28 | 3.76e−04 | 0.75 | 0.38,1.48 | 0.404 |
| Age | 0.98 | 0.97,0.99 | 5.57e−09 | 0.98 | 0.97,0.99 | 9.14e−09 |
| Sex | 1.12 | 0.94,1.32 | 0.210 | 1.12 | 0.94,1.33 | 0.192 |
| Body Mass Index (kg/m2) | 1.01 | 1.00,1.02 | 0.147 | 1.01 | 1.00,1.02 | 0.142 |
| Any Comorbidity | 1.20 | 1.01,1.41 | 0.032 | 1.19 | 1.00,1.40 | 0.045 |
| Active Smoking | 1.11 | 0.92,1.34 | 0.276 | 1.12 | 0.93,1.35 | 0.241 |
| Number of Positive SARS-CoV-2 Swabs | 1.21 | 1.11,1.32 | 1.94e−05 | 1.20 | 1.10,1.31 | 3.06e−05 |
| Patient Contact | 1.02 | 0.88,1.18 | 0.791 | 1.02 | 0.88,1.18 | 0.820 |
| Home with Children | 0.95 | 0.81, 1.11 | 0.495 | 0.95 | 0.81, 1.12 | 0.548 |
| Influenza vaccination season 2023/2024 | 0.84 | 0.71,0.99 | 0.036 | 0.84 | 0.71, 0.99 | 0.035 |

[1]*aIRR* adjusted Incidence Rate Ratio, *CI* Confidence Interval; [2]using Wald-test.
Model 1 (without time of last vaccination) and model 2 (with time of last vaccination).

**Table 3 | Negative binomial model for association of SARS-CoV-2 vaccination with number of ILI episodes after applying inverse-probability weighting**

| Characteristics | Model 1: Without Time | | | Model 2: With Time | | |
|---|---|---|---|---|---|---|
| | aIRR[1] | 95% CI[1] | p-value | aIRR[1] | 95% CI[1] | *p*-value[2] |
| Time of last Vaccine | | | | 1.26 | 1.12,1.43 | 1.80e−4 |
| Number of Vaccines | | | | | | |
| 0 | — | — | | — | — | |
| 1 or 2 | 1.27 | 1.12,1.43 | 1.56e−4 | 0.89 | 0.71, 1.12 | 0.316 |
| 3 | 1.45 | 1.29,1.63 | 7.28e−10 | 0.90 | 0.68, 1.18 | 0.447 |
| 4 | 1.52 | 1.34,1.73 | 7.96e−11 | 0.76 | 0.52, 1.12 | 0.165 |
| Influenza vaccination season 2023/2024 | 0.87 | 0.79,0.96 | 0.00630 | 0.87 | 0.79, 0.96 | 0.00647 |

[1]*aIRR* adjusted Incidence Rate Ratio, *CI* Confidence Interval, [2]using Wald-test.
Model 1 (without time of last vaccination) and model 2 (with time of last vaccination).

above, except the number of positive swabs, were successfully balanced (Supplementary Fig. 3). The results of the negative binomial models in the IPTW population with ILI as outcome were similar to those of model 2 with time of last vaccination (aIRR 1.26, 95% CI 1.12–1.43) and seasonal influenza vaccination (aIRR 0.87, 95% CI 0.79–0.96) being significantly associated with the outcome, whereas vaccination status was not (Table 3).

### Secondary outcome: loss of workdays
In multivariable analysis, SARS-CoV-2 vaccination status likewise showed an association with cumulative workdays lost due to ILI (aIRR for 1 or 2 vaccinations 1.13, 95% CI 0.80–1.58; for 3 vaccinations 1.49, 95% CI 1.08–2.01; for 4 vaccinations 1.50, 95% CI 1.04–2.13), as did the presence of comorbidities, number of positive SARS-CoV-2 swabs and BMI. Influenza vaccination and direct patient contact were associated with fewer days lost (Table 4).

### Discussion
In this investigation of factors associated with post-pandemic ILI episodes and loss of workdays, we found the number of SARS-CoV-2 vaccinations and previous SARS-CoV-2 infections to be positively associated with both outcomes. In contrast, seasonal influenza vaccination correlated with a decreased risk. For SARS-CoV-2 vaccination, the stronger association with a more recent timing of the vaccination as compared to the absolute number of vaccine doses suggests a temporal effect. This signal was consistent in the IPTW analysis and sensitivity analyses.

With more than one third of HCW experiencing ILI during the winter period and an average loss of two workdays per employee, respiratory infection constitutes a relevant burden in our cohort. SARS-CoV-2 vaccination was not associated with a protective effect against ILI. On the contrary, we observed a clinically relevant risk of ILI with up to 70% increase. This is in line with a number of recently published studies that focused on the risk of SARS-CoV-2 reinfection as outcome[7–9,18]. They reported hazard ratios between 1.4 and 3.6 for people with vs. those without (or less doses of) SARS-CoV-2 vaccine[4,7–9]. In our data, this risk association was stronger with a more recent vaccination than with the number of doses received, suggesting the association to be temporary and to wane over time.

The association of SARS-CoV-2 booster vaccination with increased risk for SARS-CoV-2 reinfection has been hypothesized to arise from selection bias occurring by analysing individuals with previous infection that might be more susceptible on average[10]. To account for these factors of observational data, the IPTW was performed with results pointing towards a

**Table 4 | Multivariable regression analysis for factors associated with workdays lost**

| Characteristics | aIRR[1] | 95% CI[1] | p-value[2] |
|---|---|---|---|
| Number of Vaccines | | | |
| 0 | — | — | |
| 1 or 2 | 1.13 | 0.80, 1.58 | 0.479 |
| 3 | 1.49 | 1.08, 2.01 | 0.011 |
| 4 | 1.50 | 1.04, 2.13 | 0.028 |
| Age | 0.99 | 0.98, 1.00 | 0.071 |
| Sex | 0.88 | 0.71, 1.11 | 0.285 |
| Body Mass Index (kg/m2) | 1.02 | 1.00, 1.04 | 0.042 |
| Any Comorbidity | 1.54 | 1.24, 1.92 | 1.14e-04 |
| Active Smoking | 1.01 | 0.79, 1.31 | 0.931 |
| Number of Positive SARS-CoV-2 Swabs | 1.27 | 1.13, 1.43 | 5.09e-05 |
| Patient Contact | 0.78 | 0.65, 0.94 | 0.010 |
| Home with Children | 0.92 | 0.75, 1.13 | 0.423 |
| Influenza vaccination season 2023/2024 | 0.74 | 0.60, 0.92 | 0.006 |

[1]aIRR adjusted Incidence Rate Ratio, CI Confidence Interval; [2]using Wald-test.

causal relation of SARS-CoV-2 vaccination with the outcomes. Additionally, the association remained significant even after adjusting for the number of previous SARS-CoV-2 infections, which might serve as a surrogate for individual susceptibility. Also, the association of seasonal influenza vaccination with decreased risk for the outcomes is in line with widely accepted evidence[19–21]. In addition, the effect size aligns with congregated evidence[19–21] supporting the validity of our finding. Various laboratory studies suggest immune imprinting to occur by SARS-CoV-2 booster vaccination; however, data on explanatory biological mechanisms are scarce. Some studies suggest that SARS-CoV-2 vaccines may have heterologous immunological effects and alter the innate immune response[22–24]. Interestingly, the variability and extent of metabolic and transcriptomic changes of innate immune cells to various stimuli have been shown to be enhanced after vector-borne SARS-CoV-2 vaccination[22], but dampened after mRNA-SARS-CoV-2 vaccination[25], which was received by the majority of our cohort. Also, alterations of the adaptive immune system have been shown in animal models with highly immunogenic lipid nanoparticles of SARS-CoV-2 mRNA vaccines[26,27], inducing T-cell exhaustion[27]. As with our results, a waning of this effect over time has been observed. Also, the protection against reinfection is known to be differentially influenced by immunity being derived from infection, vaccination or a combination of both[6,28]. Natural infection, also mirrored in higher anti-N, seems to correlate with protection[29], and the proportion of anti-N-positive individuals seen in the less vaccinated groups of our cohort might thus contribute to the association seen. Also, the application of heterologous vaccine schedules (i.e., mRNA and vector-borne vaccine formulations) has been shown to be associated with a decreased risk of COVID-19 outcomes[30]. However, further research is warranted to elucidate the presence and nature of mechanisms underlying the observable association.

Interestingly, those being involved in direct patient care reported less days of sick leave despite showing no difference in the number of ILI experienced. Earlier studies have shown that large proportions of HCW with ILI work despite being symptomatic[31,32]. While during the pandemic, no difference between those involved in patient contact compared to those without could be seen[33], post-pandemic behaviour might be more comparable to pre-pandemic habits[34] when a sense of duty towards the patients led many HCW to work despite symptoms[35,36]. This puts not only patients, but also fellow HCW at risk for infection[37–39] and may contribute to the average loss of 2 workdays per HCW during only one winter season. In line with previous reports[31,34], these numbers pose a relevant burden on healthcare systems. This emphasizes the need for the identification of strategies protecting HCW from ILI.

An important strength of our study is the availability of detailed information on weekly symptoms from a cohort with a high response and low attrition rate. Also, the relatively large group of unvaccinated individuals provides unique opportunities and presents a major strength. Vaccination status is furthermore clearly defined for each individual and although being self-reported, previous validation has shown their reliability[11,40]. However, our study does have limitations. First, we did not test for pathogens, so viral etiologies can only be extrapolated from correlation with local epidemiologic data. While HCWs might have increased exposure to respiratory pathogens due to their occupation, we did not find patient contact to be associated with the number of ILI. Second, the time of last vaccination against SARS-CoV-2 was >1 year ago for the vast majority of our cohort, while seasonal influenza vaccination could also be administered during the follow-up period. This could skew the results; however, a sensitivity analysis including only those without influenza vaccination showed no diverging results. Third, the effect of bivalent vaccines could not be included in our main model due to multicollinearity. However, according to the results of our sensitivity analysis, we did not see any additional effect of the bivalent vaccine. Fourth, we only included predominantly healthy and female HCW, which limits the generalizability to other populations, and the study population is relatively small. Fifth, although IPTW is an acknowledged and double-robust method for pseudo-randomization of observational studies, only known confounders can be accounted for, leaving the possibility of unmeasured confounding. Lastly, we can only speculate on the immunological mechanisms that may underlie these findings. Since we obtained no cell-containing samples from the participants, we were not able to further investigate those mechanisms that most probably involve cellular immunity pathways. Also, residual confounding and the limitations posed by study design might contribute to the effects observed.

Based on our data, we conclude that SARS-CoV-2 booster vaccination did not contribute to a measurable protection of the HCW workforce studied and may even temporarily increase the likelihood of symptomatic infection and workday loss. However, further research confirming our results and investigating the purported immunological mechanisms behind this phenomenon are needed.

## Data availability
Data underlying Fig. 2 are shown in Supplementary Table 3. The raw study data are not yet publicly available as some additional analyses are still pending. However, the data are available from the corresponding author upon reasonable request.

## Code availability
Statistical codes are available on the dryad data repository (https://doi.org/10.5061/dryad.v41ns1s88)[41].

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

## Acknowledgements

This work was supported by the Swiss National Science Foundation (Grant number 31CA30_196544; grant number PZ00P3_179919 to Philipp Kohler) and the Swiss Academy of Medical Sciences (Grant number YTCR 12/22 to Tamara Dörr).

## Author contributions

P.K., C.R.K., M.S. and S.P.K. were involved in acquisition of funding for and management of the cohort; P.K., C.R.K., S.P.K., and T.D. developed the concept of the study; F.G. was responsible for data management; P.K., M.A.P., J.L., S.H., T.B. and T.D. were involved in planning of statistical analysis which were conducted by J.L. and T.D.; A.C., E.K., J.C.M., M.v.K., M.R., R.S., D-V-G., C.R.K. and P.K. were responsible for data collection and participants inquiries at their respective institutions; T.D., J.L. and P.K. wrote the main manuscript. T.D., J.L., T.B., A.C., F.G., S.H., E.K., J.C.M., M.A.P., M.S., M.v.K., M.R., R.S., D-V-G., C.R.K., S.P.K. and P.K. gave critical review of and edited the manuscript.

## Competing interests

The authors declare no competing interests.

## Ethics

The study was approved by the Ethics Committee of Eastern Switzerland (#2020–00502).

## Additional information

**Peer review information** : *Communications Medicine* thanks Ya-Jankey Jagne and Eero Poukka for their contribution to the peer review of this work. [A peer review file is available].

[1]HOCH, Cantonal Hospital St.Gallen, Division of Infectious Diseases, Infection Prevention and Travel Medicine, St.Gallen, Switzerland. [2]Epidemiology, Biostatistics and Prevention Institute (EBPI), University of Zurich, Zurich, Switzerland. [3]Cantonal Hospital Graubuenden, Chur, Switzerland. [4]Geriatric Clinic St. Gallen, St. Gallen, Switzerland. [5]Center for Neurological Rehabilitation Zihlschlacht, Zihlschlacht, Switzerland. [6]Hirslanden Clinic Stephanshorn, St. Gallen, Switzerland. [7]Fuerstenland Toggenburg Hospital Group, Wil, Switzerland. [8]Hirslanden Clinic Zurich, Zurich, Switzerland. [9]Thurgau Hospital Group, Division of Infectious Diseases and Hospital Epidemiology, Muensterlingen, Switzerland. [10]Department of Research and Development, Swiss National Centre for Infection Prevention (Swissnoso), Berne, Switzerland. [11]Department of Infectious Diseases and Hospital Epidemiology, Children's Hospital of Eastern Switzerland, St. Gallen, Switzerland. [16]These authors contributed equally: Tamara Dörr, Joanne Lacy. ✉e-mail: philipp.kohler@h-och.ch

## for the SURPRISE+ Study Group

Alexia Cusini[3], Tamara Dörr ⓘ[1,16], Stephan Goppel[12], Fabian Grässli[1], Christian R. Kahlert ⓘ[1,11], Joelle Keller[8], Simone Kessler[1], Philipp Kohler ⓘ[1,2]✉, Stefan P. Kuster[1], J. Carsten Möller[5], Maja F. Müller[8], Philip Rieder[8], Lorenz Risch[13,14,15], Markus Rütti[7], Matthias Schlegel[1], Reto Stocker ⓘ[8], Matthias von Kietzell[6] & Danielle Vuichard Gysin[9,10]

[12]Psychiatry Services of the Canton of St Gallen, St Gallen, Switzerland. [13]Labormedizinisches Zentrum Dr Risch Ostschweiz AG, Buchs, Switzerland. [14]Private Universität im Fürstentum Liechtenstein, Triesen, Liechtenstein. [15]Center of Laboratory Medicine, Institute of Clinical Chemistry, University of Bern, Inselspital, Bern, Switzerland.

