## [Transparent Peer Review file · Communications Medicine]

Association of SARS-CoV-2 vaccination status with risk of influenza-like illness and loss of workdays in healthcare workers

Corresponding Author: Professor Philipp Kohler

Version 0:

Reviewer comments:

Reviewer #1

(Remarks to the Author)

In their study, Dörr et al determined the association of SARS-CoV-2 vaccination status and other risk factors with the occurrence of influenza-like respiratory illness (ILI) and workdays lost due to ILI. The study was carried out in a cohort of health care workers recruited from nine healthcare networks in Northern and Eastern Switzerland. They found that the number of SARS-CoV-2 vaccinations and previous SARS-CoV-2 infections were positively associated with the occurrence of ILL and loss of workdays. The authors concluded that their data does not support annual SARS-CoV-2 vaccination in health care workers (HCW) as SARS-CoV-2 booster vaccination does not contribute to a measurable protection of HCW workforce and may even temporarily increase the likelihood of symptomatic infection and workday loss. There is no direct link between vaccine induced events and the 'ILI' measured after some time. It is more likely that there is environmental exposure to other infections and/ or that older ages groups are more susceptible to inflammatory co-morbidities.

1. Although the statistical methods used were robust, the immunological mechanism by which SARS-CoV-2 vaccination would impact on increased incidence of influenza like illness was not fully explored.
2. The authors reported that those with 3-4 SARS- CoV-2 vaccine doses tended to be older and more likely to report comorbidities. In all the models, the adjusted incidence rate ratio of presence of any comorbidity was >1 highlighting a higher incidence of ILL and workdays lost on the HCWs with comorbidities. The presence of comorbidities can lead to increased susceptibility to infections, weakened immune system, increased chronic inflammatory disease and increased risk of severe/prolonged infections.
3. The authors did mention the type of comorbidities in the supplementary data Table S4. It would be helpful if the frequency of each comorbidity for the different groups is shown as some of these conditions can be associated with chronic inflammation and may lead to ILI symptoms.
4. The authors did acknowledge that they did not test for other pathogens. However, HCW have an increased exposure to respiratory pathogens in the hospital which may also present with ILI. Can the authors discuss this please?
5. The authors should be cautious with the strong conclusion to not recommend annual SARS-CoV-2 vaccination as majority of the HCW received the mRNA vaccine and 83% were females. This conclusion / recommendation is not supported by the data set shown and should be dropped.
6. Line 222-3 participants indicated the presence of any of 22 respiratory, gastrointestinal and general symptoms (Table S1)- This is not shown. Authors should check the data.
7. The authors showed that age was associated with a decreased risk of ILI but vaccination status with 3 and 4 doses was significantly associated with the number of ILI. Can the authors please clarify this apparent contradiction as the older age group were the ones receiving more vaccine doses.

Reviewer #2

(Remarks to the Author)

Thank you for the opportunity to review the work of Dörr et al.

During the COVID-19 pandemic, it was discussed that COVID-19 vaccinations might prevent sick leave among healthcare workers. Some have proposed this idea, although the level of evidence has not been strong.

- 1) Could the authors analyze bivalent booster recipients vs. non-recipients? I think this comparison would be more relevant

for policymaking, as current considerations focus on whether seasonal COVID-19 boosters should be recommended for healthcare workers.

2) When was the vaccination associated with an increased risk of ILI-related sick leave? If this association was particularly strong during the initial weeks after vaccination, it might suggest that the side effects of COVID-19 vaccination caused the sick leave.

3) Although I enjoyed the paragraph on immune imprinting in the discussion, I'm not fully convinced that the data in this study indicate that immune imprinting caused the increased risk related to ILI-related sick leave among vaccinated. I think residual confounding or limitations in study design (e.g., target trial emulation or an RCT might have been a superior approach) could also explain the findings. Nevertheless, the results are very interesting and should be published. I would only suggest that the limitations be more highlighted in the paragraph between lines 137–181.

4) The study population is relatively small. I suggest adding this point as an additional limitation in the discussion.

Minor comments:

5) Discussion, lines 157–159: This is a very interesting discussion! The authors could consider citing the following Nordic study that I'm aware of: <https://doi.org/10.1136/bmj-2022-074325>. In that study, heterologous vaccine schedules were associated with a decreased risk of COVID-19 outcomes compared to homologous schedules.

Version 1:

Reviewer comments:

Reviewer #1

(Remarks to the Author)

I am satisfied with the revisions on the manuscript.

Reviewer #2

(Remarks to the Author)

Thank you for the clear responses!

I have just one additional comment regarding Response 2: Did you have data on sick leave during the 0–14 days following vaccination? If so, please consider including those in the manuscript, as such data would be valuable for policymakers.

Best of luck!

Referee expertise:

Referee #1: Immunogenicity, Vaccination, human studies, Influenza like illness

Referee #2: COVID-19 vaccine studies, vaccine effectiveness, human cohort studies, observation studies

Reviewers' comments:

Reviewer #1 (Remarks to the Author):

In their study, Dörr et al determined the association of SARS-CoV-2 vaccination status and other risk factors with the occurrence of influenza-like respiratory illness (ILI) and workdays lost due to ILI. The study was carried out in a cohort of health care workers recruited from nine healthcare networks in Northern and Eastern Switzerland. They found that the number of SARS-CoV-2 vaccinations and previous SARS-CoV-2 infections were positively associated with the occurrence of ILI and loss of workdays. The authors concluded that their data does not support annual SARS-CoV-2 vaccination in health care workers (HCW) as SARS-CoV-2 booster vaccination does not contribute to a measurable protection of HCW workforce and may even temporarily increase the likelihood of symptomatic infection and workday loss. There is no direct link between vaccine induced events and the 'ILI' measured after some time. It is more likely

that there is environmental exposure to other infections and/ or that older ages groups are more susceptible to inflammatory co-morbidities.

1. Although the statistical methods used were robust, the immunological mechanism by which SARS-CoV-2 vaccination would impact on increased incidence of influenza like illness was not fully explored.

Thank you for emphasizing this point. We acknowledge that this is a major limitation of our study. As our expertise lies in epidemiological research we chose not to extensively speculate on immunological mechanisms but contribute to the current evidence by adding our epidemiological perspective. Also, as we only collected serum samples from the participants, we were not able to further investigate those mechanisms that most probably involve cellular immunity pathways. This has been added to the limitations.

2. The authors reported that those with 3-4 SARS- CoV-2 vaccine doses tended to be older and more likely to report comorbidities. In all the models, the adjusted incidence rate ratio of presence of any comorbidity was >1 highlighting a higher incidence of IIL and workdays lost on the HCWs with comorbidities. The presence of comorbidities can lead to increased susceptibility to infections, weakened immune system, increased chronic inflammatory disease and increased risk of severe/prolonged infections.

Thank you for pointing this out specifically. Also here (and in reference to our answer regarding age below), we have to assume that presence of comorbidities is independently associated with the experience of ILI. This factor has been one of the reasons why we opted for complementing the traditional multivariable analysis with an IPTW analysis. The latter is designed to statistically equalize the population as well as possible in order to diminish the effect of confounding. The IPTW analysis used

propensity scores to create a weighted sample where the distribution of measured covariates, such as comorbidities, is balanced across the different treatment groups, in this case, the treatment is the number of vaccine doses received. This ensured that comparisons between groups are not confounded by differences in comorbidities or other baseline characteristics.

In addition, influenza vaccination was associated with protection against ILI in this population, although those with more comorbidities were also more likely to receive the influenza vaccine (with 24.8% of those without comorbidities vs. 29.9% of those with comorbidities; $p=0.06$). Following the argument raised, we would expect a similar paradoxical effect of the influenza vaccine, which was definitely not the case in our analysis. Thus, we think that these points at least justify publication and careful interpretation and discussion of the results.

3. The authors did mention the type of comorbidities in the supplementary data Table S4. It would be helpful if the frequency of each comorbidity for the different groups is shown as some of these conditions can be associated with chronic inflammation and may lead to ILI symptoms.

Thank you for adding this comment. We have added the categories of different comorbidities we collected and the respective counts in Table 1. Interestingly, marked differences in frequency can be seen only in arterial hypertension as well as pulmonary disease while some other comorbidities where we would expect a higher susceptibility to infections such as immunosuppression, diabetes or cancer were equally distributed among groups.

4. The authors did acknowledge that they did not test for other pathogens. However,

HCW have an increased exposure to respiratory pathogens in the hospital which may also present with ILI. Can the authors discuss this please?

Thank you for this suggestion. We acknowledge that generalizability could be an issue here, however, in our cohort we have looked at HCW with and without direct contact to patients (including HCW working in the kitchen or IT specialists, researchers or lab personnel working in separate hospital buildings) which is closely linked to the occupational exposure to respiratory pathogens. There we have failed to find an association with the number of ILI which is why we did not elaborate on this in the discussion. We have now added an according statement.

5. The authors should be cautious with the strong conclusion to not recommend annual SARS-CoV-2 vaccination as majority of the HCW received the mRNA vaccine and 83% were females. This conclusion / recommendation is not supported by the data set shown and should be dropped.

We appreciate your feedback on this statement which may read overly confident. We have reworded the conclusions to account for the limitations imposed by the characteristics of our population, the study design, and the lack of pathogen testing. Thus, we have deleted the statement on vaccination recommendations and decided to stick to observations.

6. Line 222-3 participants indicated the presence of any of 22 respiratory, gastrointestinal and general symptoms (Table S1)- This is not shown. Authors should check the data.

Thanks for pointing that out, we have added the additional table listing the symptoms asked as it had been removed accidentally.

7. The authors showed that age was associated with a decreased risk of ILI but vaccination status with 3 and 4 doses was significantly associated with the number of ILI. Can the authors please clarify this apparent contradiction as the older age group were the ones receiving more vaccine doses.

Thank you for this comment. This is indeed a finding that seems contradictory initially and would remain so if we would have only looked at univariable analysis where those same associations can be found (the IRR being 0.99 and 95% CI 0.98-0.99 for age and 1.25 (0.40-0.63), 1.34 (1.05-1.74) and 1.41 (1.06-1.87) for the number of vaccinations received respectively). However, the purpose of the multivariable analysis was to adjust for those factors and enable us to statistically evaluate their effects independently of each other. This leads us to the conclusion that the associations found for age and vaccine doses respectively are independent from each other and that - despite the fact that older participants might be more likely to get additional doses of vaccines - increasing age alone would make one less prone to an ILI while the number of vaccine doses received offsets this effect.

Reviewer #2 (Remarks to the Author):

Thank you for the opportunity to review the work of Dörr et al.

During the COVID-19 pandemic, it was discussed that COVID-19 vaccinations might prevent sick leave among healthcare workers. Some have proposed this idea, although the level of evidence has not been strong.

1) Could the authors analyze bivalent booster recipients vs. non-recipients? I think this comparison would be more relevant for policymaking, as current considerations focus on whether seasonal COVID-19 boosters should be recommended for healthcare workers.

Thank you for this comment. This is indeed a point which we had discussed as a team during the analysis thoroughly and are happy to answer on why we chose the components of our analysis as we did. The main problem arising is the high percentage of bivalent vaccine recipients in the group with 4 vaccinations as compared to the other groups (see Table 1). This of course results in multi-collinearity with both, the total number of doses as well the timing of the last vaccination. Apart from performing univariable analyses, where number and timing of vaccination were more strongly associated with ILI than receiving bivalent vaccine formulation, we also looked at the statistical model fit in the multivariable analyses and chose the one with the superior fit. Additionally, we performed a sensitivity analysis where we combined the groups with 3 and 4 vaccinations to eliminate multi-collinearity (since only those with 4 vaccinations showed a high percentage of bivalent vaccine recipients) and got the same signals, i.e. bivalent vaccination was associated with increased risk of ILI, but only as long as the variable “time since vaccination” was not included. Therefore, we believe that the main factor contributing to this effect was the fact that bivalent vaccines were given more recently than the univalent vaccines. To clarify that in the manuscript we have added this sensitivity analysis in the supplementary material and amended the manuscript accordingly.

2) When was the vaccination associated with an increased risk of ILI-related sick leave?

If this association was particularly strong during the initial weeks after vaccination, it might suggest that the side effects of COVID-19 vaccination caused the sick leave.

Thank you for pointing this out. This is indeed correct, however, we excluded those ILI that occurred in the first 14 days after SARS-CoV-2 vaccination to account for that. We have reworded this part in the methods section to state it more clearly.

3) Although I enjoyed the paragraph on immune imprinting in the discussion, I'm not fully convinced that the data in this study indicate that immune imprinting caused the increased risk related to ILI-related sick leave among vaccinated. I think residual confounding or limitations in study design (e.g., target trial emulation or an RCT might have been a superior approach) could also explain the findings. Nevertheless, the results are very interesting and should be published. I would only suggest that the limitations be more highlighted in the paragraph between lines 137–181.

Thank you for this feedback. We also think that it is very interesting but wanted to limit our speculations on that since we do not have expertise in immunological research.

Target trial design would have been desirable, however, this was not feasible given the sample size of our study. We have added some details and reworded the respective part in the section on strengths and limitations to emphasize this.

4) The study population is relatively small. I suggest adding this point as an additional limitation in the discussion.

Thank you. We have added this to the limitations' section.

Minor comments:

5) Discussion, lines 157–159: This is a very interesting discussion! The authors could

consider citing the following Nordic study that I'm aware of: <https://doi.org/10.1136/bmj-2022-074325>. In that study, heterologous vaccine schedules were associated with a decreased risk of COVID-19 outcomes compared to homologous schedules.

Thank you for this suggestion. This study is indeed very interesting and we have added this element in the discussion and included the reference. However, we were not able to study this effects in our cohort since the large majority had received mRNA vaccines in a homologous fashion.

Referee expertise:

Referee #1: Immunogenicity, Vaccination, human studies, Influenza like illness

Referee #2: COVID-19 vaccine studies, vaccine effectiveness, human cohort studies, observation studies

REVIEWERS' COMMENTS:

Reviewer #1 (Remarks to the Author):

I am satisfied with the revisions on the manuscript.

Thank you.

Reviewer #2 (Remarks to the Author):

Thank you for the clear responses!

I have just one additional comment regarding Response 2: Did you have data on sick leave during the 0–14 days following vaccination? If so, please consider including those in the manuscript, as such data would be valuable for policymakers.

Best of luck!

Thank you for your comment. We revisited our data and identified 57 individuals who received a SARS-CoV-2 vaccine during the whole follow-up period. As described in the manuscript, these individuals were excluded from the final analysis.

Among these 57 individuals, 4 reported missing work within 14 days post-vaccination, with a mean of 1.25 days of sick leave. Using the same definition of ILI symptoms applied in the manuscript, we found that 3 individuals that reported ILI symptoms within 14 days.

Given the very small sample size and limited number of events, we believe these data are insufficient to draw meaningful conclusions and may be potentially misleading if included in the manuscript. We therefore suggest not incorporating these results into the manuscript. We hope that reviewer 2 and the editor agree.